# The relationship between short video usage and academic achievement among elementary school students: The mediating effect of attention and the moderating effect of parental short video usage

Qiong Gong[1,2], Ting Tao[1,2]*

1 Institute of Psychology, Chinese Academy of Sciences, Beijing, China, 2 Departement of Psychology, University of Chinese Academy of Sciences, Beijing, China

* taot@psych.ac.cn

## Abstract

Short videos have gained widespread popularity among elementary school students in China. As a form of entertainment media, their usage has steadily increased among adolescents in recent years. This phenomenon has sparked extensive discussions in society, especially against Chinese parents' high concern for their children's academic performance. Therefore, this study collected 1052 valid questionnaires from elementary school students, attempting to explore the possibility that their short video usage might negatively impact their academic performance. Besides, the mechanism of this relationship was also examined from the perspective of children's attention and environmental factors of parents' short video usage. The research findings indicate that the more elementary school students use short videos, the lower their academic performance, with attention mediating in this relationship. The longer the parental short video usage duration, the exacerbating effect it has on elementary school students' negative impact on attention caused by short video usage due to its positive moderating effect. This study provides crucial insights for parents, educators, and short video platforms, offering valuable references for formulating more scientifically and logically grounded educational strategies.

**Data Availability Statement:** All relevant data are within the manuscript and its Supporting Information files.

## 1. Introduction

Short videos provide visual stimulation to the audience through techniques such as sound, light, and images [1]. This multimedia format is widely popular and has deeply integrated into various aspects of social life [2]. Short video applications like TikTok and Kwai have gained prominence in the wave of mobile internet and have become popular choices among young users globally [3]. Users of these applications typically watch and create short videos for entertainment and information while actively participating in social interactions [4]. The characteristics of short video apps, such as their brevity, mobility, entertainment value, vertical format,

**Funding:** The author(s) received no specific funding for this work. This research was supported by the National Natural Science Foundation of China under Grant No.62107038.

**Competing interests:** The authors have declared that no competing interests exist.

and interactivity, cater perfectly to the social needs of various audiences [5]. In China, the user base of short video apps has reached 962 hundreds of million, accounting for 91.5% of the overall Internet users [6]. Of particular concern is that minors, especially elementary school students, have become significant users of short video apps. The "Youth Blue Book: China Minor Internet Usage Report indicates that watching videos has surpassed playing games and listening to music, ranking first among internet activities for minors, accounting for 47.5%. The proportion of minors using short video apps like Kwai and TikTok is as high as 65.3%, making short videos an essential platform for their online life. Statistics show that the usage of short video apps among elementary school students has reached 66%, even higher than that of high school students (62.7%) [7].

Accompanying minors' widespread use of short videos is a series of concerns, especially in education. These concerns include addiction to short videos, exposure to inappropriate values, implicit sexual content, violence, risks of personal privacy exposure, and premature adult-oriented tendencies [8]. In China, due to the intense academic competition and pressure during elementary school, educators and parents are particularly concerned about the potential negative impact of social media on students' academic performance [9]. In the elementary school stage, students are at a critical period of earning and growth, where they are establishing study habits and academic foundations, which will serve as the cornerstone for future academic success. However, the use of short videos is often accompanied by long screen time, which may lead to a reduction in study time and even addiction [10].

Prior research has explored chiefly the psychological mechanisms through which media usage negatively affects academic performance from the perspectives of internet addiction or social media usage [11–13]. Some scholars have examined the appeal of social media to minors from the perspective of attachment, suggesting that minors form emotional connections and care when interacting with online communities [3, 14]. In comparison to the extensive research on the effects of "addiction" and "attachment" on minors, further research is needed to deepen the understanding of how specific media like short videos may influence minors. Additionally, the underlying conditions of psychological activities before the influence of short video usage on academic performance need to be further explored. Excessive media usage may lead to various learning issues, such as attention deficits, poor time management, and reduced study time [3].

Given that short videos are rich in visual and auditory elements and can convey a vast amount of information in a short time frame. The study explores how short videos might affect minors' "attention," which is a constant psychological process, spans, and academic performance, based on the Attention Restoration Theory(ART). It suggests that the appealing content of such videos could lead to attention deficits, poor time management, and less study time, impacting their academic results.

The Ecological Systems Theory from Bronfenbrenner [15] posits that the family is one of the most direct and micro-level systems influencing individual development. As a "microsystem," the family plays a pivotal role in the growth and development of children, making it a crucial component of factors sensitive to social differences [16]. Chinese parents typically have high expectations for their children's academic success and tend to control or regulate their media usage [17]. However, compared to restrictive control and intervention, parents' actions and attitudes at home may profoundly influence their children [18]. For example, parents' media behaviors and attitudes toward their children's media usage can significantly shape their values and behavioral guidelines [19]. Likewise, parents' usage duration of short videos at home may subtly affect their children's media usage habits and time allocation [9]. The study aims to investigate whether the attentional processes influenced by the use of short videos could be a significant internal factor contributing to the negative impact on academic performance among minors.

## 2. Theoretical basis of the research hypothesis

### 2.1 Short video usage and academic performance

Children are exposed to various forms of screen time and electronic device usage in today's digital age. Research indicates that the use of electronic devices is negatively correlated with academic performance, regardless of gender, age, or family background [20]. A systematic review and meta-analysis covering 480,479 samples revealed a negative correlation between children and adolescents' television viewing, video game playing, and academic performance. As the time spent by children and adolescents watching TV and playing video games increases, their overall grades, language scores, and math scores tend to decrease [20]. Likewise, studies indicate that excessive use of short videos can impact students' engagement negatively behaviorally but positively emotionally and cognitively [21, 22].

Elementary school students are in the third stage, the "Concrete Operational Stage," according to Piaget's Cognitive Development Theory. In this stage, children can engage in logical thinking and operations, but these are based on specific objects and events they observe and interact with [23]. The time displacement hypothesis suggests that spending more time on television and electronic games reduces the time children spend on other activities beneficial for cognitive development, such as physical exercise, verbal communication, studying, or sleep. This reduction in cognitive effort may impact academic performance [10]. This aligns with the findings of previous studies that excessive screen time, including mobile device usage, might lead to poorer academic performance in children [24]. This explanation corresponds to the perspective that excessive screen time, including the use of short videos, can negatively impact academic performance. Short videos are often designed with highly appealing visuals and fast-paced, engaging content, which may lead to addiction among young users [25]. This addictive nature can result in prolonged engagement, potentially diverting students' time and energy from studying. As students spend more time watching short videos, the time available for completing assignments and studying might decrease [26].

The Cognitive Load Theory suggests that learning can be hindered when individuals are simultaneously presented with excessive information [27]. Videos often convey information rapidly, demanding viewers to process a large amount of content within a short period. For elementary school students whose cognitive abilities are still developing, excessive exposure to such stimulating content may lead to an overload of cognitive load, subsequently impairing academic performance [28]. Some scholars explain the relationship between short video usage and academic performance from the perspective of media multitasking. Media multitasking refers to engaging in multiple media activities simultaneously, such as watching short videos while sending comments or socializing [29]. Individuals frequently involved in media multitasking may exhibit lower cognitive control and reduced attentional abilities [30]. This implies that users of short videos, especially elementary school students, might experience compromised cognitive control if they frequently switch between learning tasks and watching short videos, affecting their academic performance.

Furthermore, the potential impact of short video usage on academic performance might be influenced by the type of content consumed. Although some short videos have educational value, underage individuals cannot judge the value of the content provided in short videos. Most short videos primarily focus on pure entertainment, which can disrupt underage individuals' mental faculties [9, 31–33]. In conclusion, this paper proposes a hypothesis.

### 2.2 Short video usage, attention, and academic performance

The Attention Restoration Theory (ART) proposed by Kaplan and Kaplan [34] suggests that natural environments can restore cognitive functions, including attention, which might be

depleted after prolonged focused attention (such as focusing on screens). While ART primarily relates to the therapeutic benefits of natural environments, it provides an insightful conceptual framework for extrapolating this theory to the context of short video usage. Since natural environments can restore attention, excessive use of short videos, with their captivating features, essentially depletes students' attentional resources, continuously diverting their focus to screen content. Previous research has provided direct evidence linking screen media use to increased attention issues in childhood, such as children spending more than two hours daily watching TV or playing screen games more likely to have concentration problems [33]. The use of digital entertainment products has been positively correlated with Attention Deficit Hyperactivity Disorder (ADHD) [35].

Additionally, Lang's Limited Capacity Model of Motivated Mediated Message Processing (LC4MP) [36] indirectly provides clues for understanding the potential risks of short video usage on students' attention to academic tasks. This model posits that individuals have limited cognitive capacity for processing information, influenced by media content characteristics and individual motivation. In short video usage, the LC4MP model can be understood as short videos being designed to be highly attractive and engaging, diverting students' attention excessively towards watching the videos and subsequently neglecting academic tasks, thus scattering their attention. The model further suggests that when media content triggers viewers' motivation, they invest more cognitive resources, potentially causing more significant interference with students' academic tasks. Therefore, the allure of short videos and students' usage behavior might be critical factors affecting their concentration.

Attention difficulties impact the academic performance of children and adolescents, which is a significant concern in education and psychology. As an illustration, ADHD patients typically struggle to maintain focus for extended periods and are easily distracted by external stimuli, severely affecting their academic achievements [37]. Attention difficulties might lead to students struggling to maintain focus, being easily distracted, and consequently spending more time and effort on tasks. This group of students must invest more time and energy than their peers to complete academic tasks, adversely affecting their academic performance [38]. Yang and Chen's study [39] indicated that students with ADHD often exhibit lower academic scores and poorer learning skills, as they cannot fully engage in classroom activities and easily miss essential teaching content, resulting in comparatively lower levels of knowledge and understanding. This affects their exam scores and can lead to higher dropout rates. Moreover, Short video addiction is linked to avoiding learning and lacking commitment, while study avoidance is associated with silent classroom behavior. Parents and educators should be aware of the negative impact of short video addiction on learning behavior [21].

The proliferation of short video usage poses new challenges to the attention of children and adolescents. The diverse and ever-changing visual and auditory stimuli on short video platforms constantly pull individuals' attention. Anderson and Dill [40] highlighted the attentional drain caused by electronic media, where fast-paced stimuli and sustained interactivity in electronic games can lead to players' attention being scattered, making it difficult to focus on other tasks for extended periods. This situation also likely exists in short videos since they often possess similar stimulating properties. From the perspective of multitasking, Guo and Liu's study [41] found that content on electronic devices usually requires more frequent attention switches than traditional media. This frequent switching might train users to shift their attention rapidly among multiple information sources. Still, it could also decrease the ability to sustain singular-task attention for prolonged periods. This is challenging for learning tasks since students typically need to concentrate on a single subject or task for extended periods. Students who excessively use electronic devices are more prone to attention problems. Moreover, over time, these students become increasingly reliant on short videos, making it more challenging to

focus on tasks requiring concentrated attention [42]. Thus, short video usage might adversely affect individuals' attention, and those who frequently use electronic media tend to perform poorly on tasks requiring focused attention in their daily lives [21, 43]. In summary, existing studies provide preliminary evidence supporting the relationship between short video usage, attention, and academic performance.

## 2.3 Parental short video usage duration as a moderator from a family environment perspective

According to the Social Cognitive Theory, environmental and cognitive factors often influence a child's behavior [44]. Nie, Zheng, and Zhang [45] emphasized the importance of the family environment, encompassing both physical aspects (such as home arrangement and atmosphere) and psychological factors (such as family rules and cultural values), in a child's growth. Research has demonstrated the significant impact of the family environment on a child's media usage behavior [46]. Parental media usage behavior may influence a child's media habits and attention through its impact on the family environment. Specifically, when parents frequently use short-form videos, children might imitate their behavior and develop similar usage patterns [44]. Moreover, parents' attitudes toward media usage may subtly shape children's media behavior. Suppose children frequently observe their parents using short-form video apps at home. In that case, they may perceive it as a standard and acceptable behavior, leading to its normalization within the family values [47]. In a family-oriented context, parental behaviors and attitudes will demonstrate a modeling effect on a child's behavior [48, 49]. Researchers like Jordan et al. [50] and Hefner et al. [51] have shown that parents' own media consumption significantly influences children's media usage duration, highlighting the impactful modeling effect of parental media behavior.

Parental media usage patterns might negatively impact a child's attention. When parents frequently engage with social media applications, their attention might be diverted, affecting their ability to concentrate during interactions with their children. This diversion could indirectly lead to the child's reduced attention during parent-child interactions [52, 53]. Parents may use mobile media as an entertainment tool during quality time with their children, reducing interactive opportunities and missing chances to enhance the child's attention levels [47]. Therefore, this study incorporates parental short video usage duration to investigate and refine the influence and mechanisms between elementary school students' short video usage and attention.

## 2.4 Research hypothesis

*H1*: *The higher the level of short video usage among elementary school students, the lower their academic performance.*

*H2*: *The higher the level of short video usage among elementary school students, the weaker their attention.*

*H3*: *Attention plays a mediating role in the relationship between short video usage and academic performance.*

*H4*: *Parental short video usage duration positively moderates the relationship between elementary school students' short video usage and attention, exacerbating the negative impact of students' short video usage on their attention and further affecting their academic performance.*

This study proposes the following research model in light of the literature and hypotheses above (See Fig 1).

## 3. Methods

### 3.1 Questionnaire design and variable measurement

This study requires collecting data from four sources: (1) students' academic performance in the second semester after the 2022~2023 winter vacation; (2) the "Questionnaire on Short Video Usage by students"; (3) Attention Test; and (4) the "Questionnaire on Short Video Usage by parents." Regarding students, the primary focus is gathering information about their short video usage and attention test results. As for parents, the data collected encompasses their awareness of their child's short video habits, their attitudes toward the child's short video usage, and their short video usage behaviors. The Institutional Review Board of Chiese Academy of Science reviewed and approved the studies involving human participants. The participants provided their written informed consent to participate in this study.

**3.1.1 Independent variable: Short video usage.**   Regarding the short video usage, this research designed 14 questions in two dimensions: 6 questions in the objective usage dimension and eight in the subjective usage dimension. The scores for all questions were totaled as an indicator of "short video usage".

Firstly, the study designed the questionnaire based on the referable items from the "Research Report on Internet Usage among Minors in China 2021" [54] to understand the current objective of short video usage among elementary school students. These questions aimed to provide a rough understanding of the frequency and duration of short video usage among elementary school students. To avoid the randomness introduced by short video usage within a specific period, the questionnaire asked about the average daily duration and frequency of short video usage during the winter vacation, as well as the duration and frequency of short video usage on weekdays and weekends during the semester before the winter vacation. For example, questions like "During the winter vacation, how long did you use short videos on average each day?" and "During the winter vacation, how many times did you use short videos on average each day?" were included, totaling six items under the dimension of "Objective short video usage." Each question was scored on a scale from 0 to 5, where 0 represented "I haven't been using short videos during this period." As the duration and frequency of short video usage increased, the scores also increased accordingly.

Elphinston and Noller [55] created an assessment tool called the "Facebook Intrusion Questionnaire," consisting of 8 questions. The questions were derived from Brown's [56] research on behavioral addiction and Walsh, White, and Young's [57] "Mobile Phones Involvement Questionnaire." Later, Hou et al. [58] translated this questionnaire into Chinese. To adapt it for this study, the researchers maintained the original question descriptions but replaced the term "Weibo" with "short video" (e.g., "I often think about short videos when I am not using them," and so on). Eight items were rated on a scale from 0 to 5, where 0 represented "Completely incompatible," and 5 represented "Completely compatible".

The scale's reliability was assessed, and Cronbach's α coefficient was found to be .90, indicating a high level of consistency among the items. Since the variable of short video usage was measured based on two subgroups, confirmatory factor analysis (CFA) was conducted using maximum likelihood estimation to assess its factor structure and validity. CFA is a commonly used statistical tool in psychology as it can test relationships between observed variables and latent variables or factors [59]. Two first-order constructs were included for the second-order construct of short video usage (α = .900). Two latent variables, objective short video usage (α = .904) and subjective short video usage (α = .896), were related to the 14 short video usage scale items, which served as observed variables. All items had relatively high factor loadings ($> .40$) on their respective latent variables. The results of confirmatory factor analysis indicated good

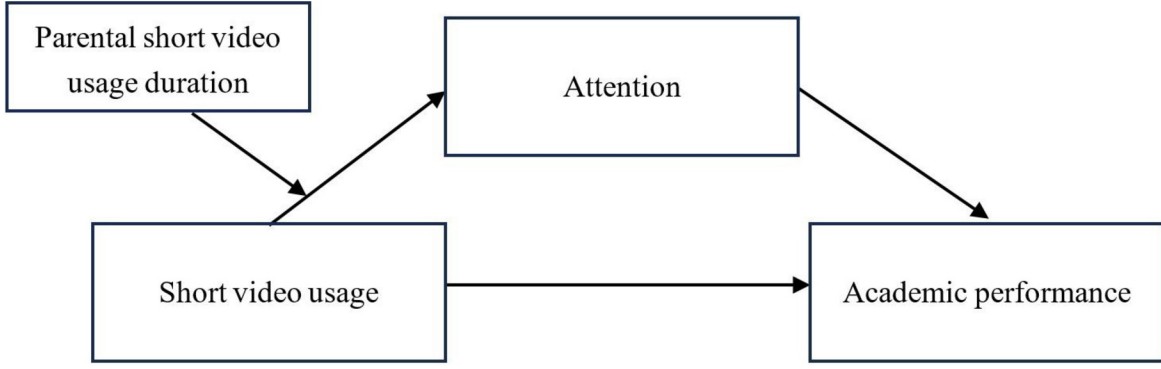

**Fig 1. Research hypotheses Model 1.**

structural validity for this scale: $\chi^2$/df = 4.897, p = .000, CFI = .962, RMSEA = .061, TLI = .955 (See Fig 2).

**3.1.2 Dependent variable: Academic performance.** The proxy variable for academic performance is students' exam scores. The researcher obtained the opening exam scores for 1052 students in the second semester after the 2022~2023 winter vacation from the academic affairs offices of two elementary schools. The exam contains three main subjects: language, mathematics, and English. Except for the mathematics test, which was out of 120 points, all other tests were out of 100 points.The scores from the three subjects are added and summed, ranging from 0–320, with higher scores indicating better academic performance.

**3.1.3 Mediator: Attention.** This test utilized the "Adolescent Attention Test" developed by Yin [59] to measure the variable, attention. This test is a timed assessment comprising a total of four sub-tests. These sub-tests include Test 1, which measures attention breadth (Circle Selection Test); Test 2, assessing attention stability (Visual Tracking Test); Test 3, evaluating attention allocation ability (Shape Recognition Test); and Test 4, measuring attention shifting ability (Addition and Subtraction Test). The qualities of attention, including breadth, stability, allocation, and shifting, constitute the overall attentional capacity [59]. After evaluating the number of correct answers in each sub-test separately, the score of variable attention is the sum of correct answers from all four sub-tests, ranging from 0 to 555. the higher the score, the better the attention.

**3.1.4 Moderator: Parental short video usage duration.** The variable parental short video usage duration measurement relies on the response to the "Questionnaire on Short Video Usage by Parents "question: "On average, how much time do you spend using short videos each day?" The variable is scored on a scale from 0 to 5, where 0 represents "Did not use short videos at all," 1 represents "0–30 minutes," and 5 represents "Above 3hr".

**3.1.5 Control variable.** Considering that other non-core variables might influence the dependent variable, this study selected four control variables: school type (urban, suburban), student grade (lower, middle, high grades), student gender (male, female), parents' education (elementary school, doctorate), of which "parent's education" refers to the one who filled out the questionnaire. The specific variable selection and measurement criteria are shown in Table 1.

## 3.2 Participants and procedure

This the study employed a stratified cluster sampling method, selecting one elementary school from both urban and suburban areas between May to july 2021 in Shenzhen, China. The

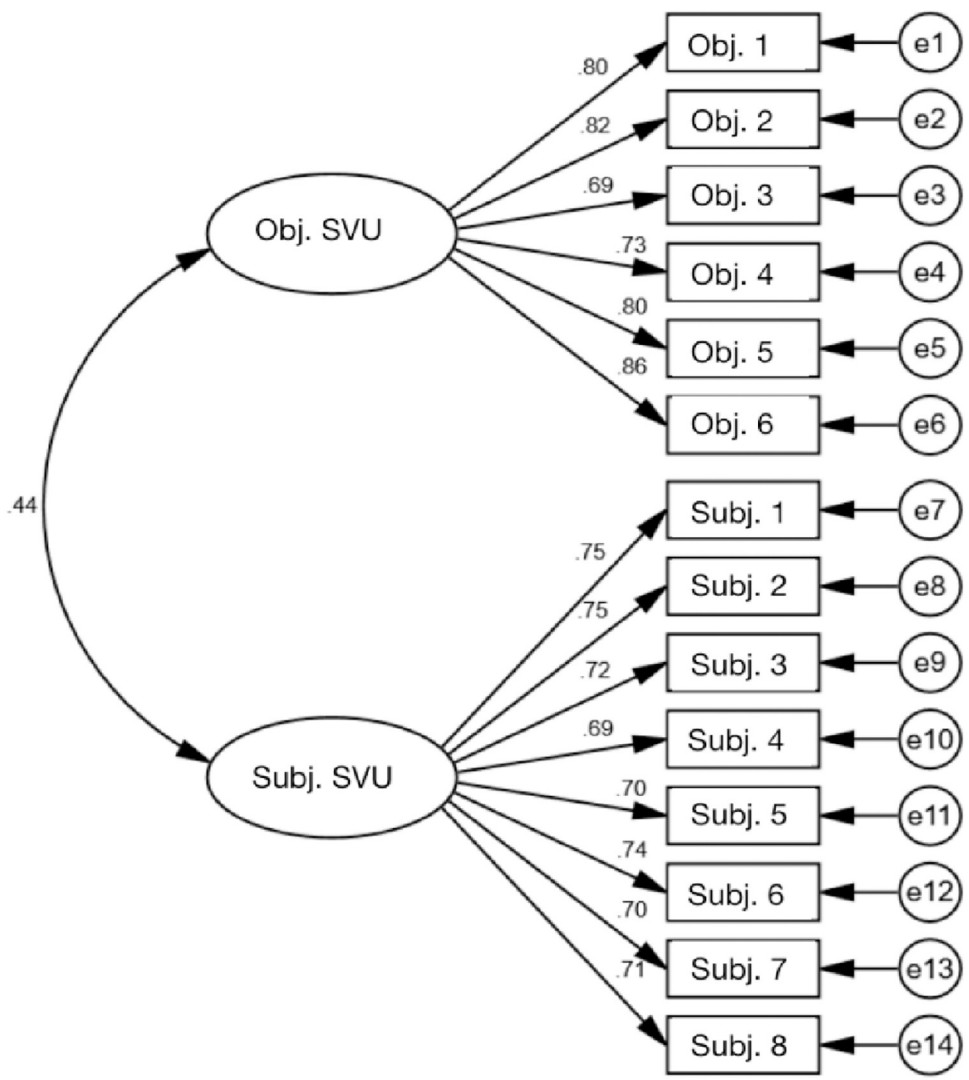

**Fig 2. Validation factor analysis results.**

suburban elementary school is where the author works, and the urban school is a sister school within the same group. Fourteen classes were selected from each school, totaling 1201 participants. Before the research commenced, a video was created to communicate the study's objectives, data utilization, and ethical guidelines to students and parents. The study received ethical clearance from the "Institute of Psychology, Chinese Academy of Sciences" on March 15, 2021.

**Table 1. Control variable.**

| Code | Variable Type | Description |
|---|---|---|
| School type | Binary variable | Urban school = 1, Suburban school = 2 |
| Students' grade | Ordered categorical variable | Low Grade = 1, Middle Grade = 2, High Grade = 3 |
| Students' gender | Binary variable | Male = 1, Female = 2, |
| Parents' education | Ordered categorical variable | Scored from 1 to 7<br>e.g., Elementary School = 1, Doctorate = 7 |

**Table 2. Sample description.**

| Category | Indicator | Frequency | Percentage (%) |
|---|---|---|---|
| School | Urban | 647 | 61.5 |
| | Suburban | 405 | 38.5 |
| Students' grade | Low Grade (1–2) | 371 | 35.3 |
| | Middle Grade (3–4) | 301 | 28.6 |
| | High Grade (5–6) | 380 | 36.1 |
| Students' gender | Male | 554 | 52.7 |
| | Female | 498 | 47.3 |
| Parents' education (the one who filled the questionnaire) | Elementary | 17 | 1.6 |
| | Junior High | 73 | 6.9 |
| | High School/Technical | 145 | 13.8 |
| | Associate degree | 338 | 32.1 |
| | Bachelor's Degree | 419 | 39.8 |
| | Master's Degree | 57 | 5.4 |
| | Doctorate | 3 | 0.3 |

Subsequently, after securing informed consent from students, parents, and teachers, the research proceeded.

After removing the problematic data, 1,052 students had all four copies of the data flush, and they were used as the study population, indicating a high-quality and representative study with an effective response rate of 87.59%. Due to the different grade structures of the two schools (urban schools had grades 1–6, while suburban schools had grades 1–4), the sample size of urban elementary schools occupied a higher proportion (61.5%). The specific sample distribution is shown in the Table 2 below.

After winter vacation, during the first week of the new semester, we distributed the"Questionnaire on Short Video Usage by Students" and the "Questionnaire on Short Video Usage by Parents" through the online platform "Wenjuanxing." Students completed it independently or with the help of parents on their cell phones at home (For students in grades 1–2 with limited literacy and comprehension skills, parents read and explained the questions or answered on their behalf).

During the second week of the new semester, we tested students' attention to the survey classes in two schools, each during a single class period, and was proctored by the class teachers. To solve the problem of teachers who had not received professional training in the administration of the attention test, the author recorded in advance a video guide for the administration of the test, in which clear instructions and detailed requirements for answering the questions were recorded in accordance with the timeline and timers of each test was set inside so that the teachers needed to play the video in front of the classroom. During the third week of the new semester, the opening exams were held among two elementary schools. Then, we got the scores from the academic affairs offices.

### 3.3 Analytical procedure

This study explores how short video usage of elementary school students affects their academic performance, including direct effects and attention's mediating role. Additionally, it examines the influence of the duration of parental short video usage on this relationship. The relationship explored in this study involves a moderated mediation model, where short video usage, attention, and academic performance could be viewed as three latent variables composed of multiple items. Hence, structural equation modeling (SEM) can be employed to test the direct

and mediating paths within the model for rigorous analysis [25]. However, the duration of parental short video usage is a continuous manifest variable and is not composed of scale items. Following the method for analyzing the moderating effects of manifest variables proposed by Fang et al. [60], regression analysis is used to test the moderating effects of manifest variables for single-level data or integrated models with moderation and mediation, as SEM is not applicable. This study will validate the mediating and moderating effects using different step-by-step tools, as shown below.

On the one hand, to examine the direct impact of elementary school students' short video usage on academic performance and the mediating role of attention within the model, the study will utilize AMOS 26.0 to construct and analyze the structural equation model (SEM). This model will incorporate several key concepts. Within the SEM, both the direct and indirect paths (via attention mediation) of short video usage on academic performance will be simultaneously estimated to test hypotheses H1, H2, and H3. The model fit will also be assessed, and various statistical tests were conducted to confirm the model's validity.

On the other hand, considering parental short video usage duration as a continuous manifest variable, this study will separately verify the moderating effects of short video usage duration on the relationship between short video usage and attention in the front path of the mediation model. The analysis will utilize Model 7 from the Process Macro 3.5 plugin in SPSS 25.0 to verify the moderating effects [61]. The results will be determined by observing the interaction terms between the independent and moderating variables.

## 4. Results

### 4.1 Descriptive and correlation analysis for core variables

Table 3 presents each variable's mean, standard deviation, minimum, and maximum values. While, Table 4 shows the statistical correlation between the operated varables.

### 4.2 Direct effect and mediating effect test

**4.2.1 Model fit test.** A Structural Equation Model (SEM) was constructed and thoroughly analyzed to explore the relationships among elementary school students' short video usage, attention, and academic performance. The fit indices of the model demonstrated a strong fit between the proposed model and the observed data ($\chi^2$/df = 2.989, GFI = 0.952, AGFI = 0.940, NFI = 0.953, TLI = 0.963, CFI = 0.968, RMSEA = 0.044). These indices met established standards ($\chi^2$/df < 3, GFI > 0.8, AGFI > 0.8, NFI > 0.9, TLI > 0.9, CFI > 0.9, RMSEA < 0.08), confirming the accuracy of the model in representing the relationships between variables (please refer to Fig 3 and Table 5).

**4.2.2 Model path test.** In terms of path coefficients, short video usage is negatively related to academic performance (standard path coefficient = -0.205, critical ratio = -4.096, p < 0.001), indicating that frequent use of short videos may have detrimental effects on academic performance. Additionally, short video usage is negatively related to attention (standard path coefficient = -0.211, critical ratio = -4.335, p < 0.001), suggesting that attention levels

**Table 3. Descriptive statistics for core variables.**

| Variable | M±SD | Min | Max |
|---|---|---|---|
| Short video usage | 38.09±12.44 | 6.00 | 64.00 |
| Attention | 192.10±107.56 | 1.00 | 539.00 |
| Academic performance | 261.73±34.342 | 42.00 | 311.00 |
| Parental short video usage | 3.01±1.00 | 0.00 | 5.00 |

**Table 4. Correlation analysis.**

| Variable | 1 | 2 | 3 | 4 |
|---|---|---|---|---|
| 1 Short video usage | 1 | | | |
| 2 Attention | -.154** | 1 | | |
| 3 Academic performance | -.271** | .221** | 1 | |
| 4 Parental short video usage | 0.021 | .113** | -0.004 | 1 |

Note:* p<0.05

** p<0.01

simultaneously decrease as short video usage increases. Furthermore, attention positively impacts academic performance (standard path coefficient = 0.202, critical ratio = 5.546, p < 0.001), demonstrating a positive relationship between good attention levels and improved academic performance.

Using the mediating effects test, Table 6 explores the further impact of short video usage on academic performance through its influence on attention. Overall, the total effect of short video usage on academic performance was -0.248 (95% confidence interval [-0.327, -0.154], p < 0.001). Specifically, short video usage had a direct and significantly negative impact on academic performance (effect size = -0.205, 95% confidence interval [-0.285, -0.109], p < 0.001), while its influence on attention resulted in a significant indirect effect on academic performance (effect size = -0.043, 95% confidence interval [-0.065, -0.022], p < 0.001). The presence of this indirect effect indicates that attention plays a partial mediating role between short video usage and academic performance. Short video usage diminishes students' attention, consequently reducing their academic performance. This suggests that excessive immersion of elementary school students in short videos might reduce their attention levels. When students are unable to concentrate, their academic performance is adversely affected.

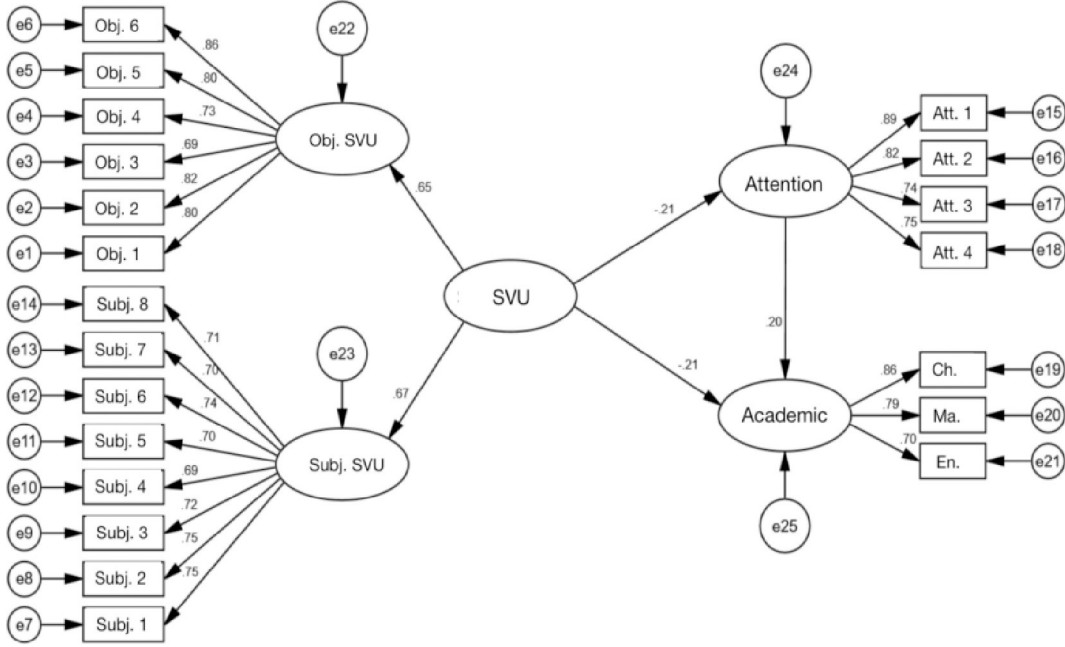

**Fig 3. Validation factor analysis results.** Note: SVU = Short Video Usage; Obj.SVU = Objective Short Video Usage; Subj. SVU = Subjective Short Video Usage; Att. = Attention.

**Table 5. Model fit indices.**

| Indices | X2/df | GFI | AGFI | NFI | TLI | CFI | RMSEA |
|---|---|---|---|---|---|---|---|
| Statistical Value | 2.989 | 0.952 | 0.94 | 0.953 | 0.963 | 0.968 | 0.044 |
| Reference Value | <3 | >0.8 | >0.8 | >0.9 | >0.9 | >0.9 | <0.08 |
| Compliance Status | Good | Good | Good | Good | Good | Good | Good |

### 4.3 Moderating effect test

This study employed Model-7 from SPSS 25.0 and its plugin Process Macro 3.5 to examine the moderating effects of parental short video usage duration separately. This analysis aimed further to explore the outcomes of the moderated mediation model. When parental short video usage duration was a moderating variable in the model, this study revealed a significant negative correlation between short video usage and attention ($\beta = -0.071$, $t = -3.792$, $p < 0.001$). Further interpretation of the interaction term indicated that parental short video usage duration moderated the relationship between short video usage and attention ($\beta = -0.092$, $t = -5.293$, $p < 0.001$). It was a positive moderation, meaning that longer parental short video usage duration enhances the negative impact of children's short video usage on their attention. In addition, when the duration of parental short video usage increased, the moderated mediation effect was reduced (effect size = -0.019). This implies that parental short video usage duration not only directly affects the relationship between students' short video usage and attention but also exacerbates the impact of short video usage on attention issues. This intensification diminishes the positive effect of attention on students' academic performance, negatively impacting their academic performance. (Please refer to Table 7 and Fig 4). Moderation Test of Parental Short Video Usage Duration in the Mediation Model).

To further understand the moderating effect of parental short video usage duration in the model, parental short video usage duration was grouped into high and low categories based on one standard deviation above and below the mean. Simple slope analyses were conducted for the interaction term. The graph shows that when parental short video usage duration is low, the predictive effect of short video usage on attention is significant (simple slope = -.071, $t = -3.811$, $p < .001$). Similarly, when parental short video usage duration is high, the predictive effect of short video usage on attention remains significant (simple slope = -.163, $t = -8.719$, $p < .001$). The slope values show an increasing trend. In other words, strengthening parental short video usage duration will intensify the negative impact of short video usage on attention compared to the low usage group.

## 5. Discussion

### 5.1 Exploring the interplay of short video usage, attention, and academic performance among elementary school students: The role of parental influence

This study aims to understand the complex relationships between elementary school students' short video usage, attention, and academic performance. Additionally, it explores the

**Table 6. Mediating effects test.**

| Mediation Path | Effect Size | Lower Bound | Upper Bound | p |
|---|---|---|---|---|
| Short Video Usage—Attention—Academic Performance (Indirect Effect) | -0.043 | -0.065 | -0.022 | 0.000 |
| Short Video Usage—Academic Performance (Direct Effect) | -0.205 | -0.285 | -0.109 | 0.000 |
| Short Video Usage—Academic Performance (Total Effect) | -0.248 | -0.327 | -0.154 | 0.000 |

**Table 7. Moderation test of parental short video usage duration in the mediation model.**

**Results for Moderation Analysis**

| Dependent Variable | Predictor Variable | $R^2$ | F | β | T | Boot LLCI | Boot ULCI |
|---|---|---|---|---|---|---|---|
| Attention | Short Video Usage | 0.804 | 272.550 | -0.071*** | -3.792 | -0.107 | -0.034 |
| | Parental Short Video Usage Duration | | | 0.079*** | 4.260 | -0.043 | 0.115 |
| | Short Video Usage × Parental Short Video Usage Duration | | | -0.092*** | -5.293 | -0.126 | -0.058 |
| | School | | | -0.169*** | -7.044 | -0.216 | -0.122 |
| | Students' grade | | | 0.658*** | 29.287 | 0.614 | 0.702 |
| | Students' gender | | | -0.007 | -0.377 | -0.043 | 0.029 |
| | Parents' education | | | 0.033 | 1.666 | -0.006 | 0.072 |
| Academic Performance | Short Video Usage | 0.506 | 0.256 | -0.106*** | -3.917 | -0.160 | -0.053 |
| | Attention | | | 0.203*** | 4.633 | 0.117 | 0.289 |
| | School | | | -0.389*** | -10.974 | -0.459 | -0.320 |
| | Students' grade | | | -0.312*** | -7.136 | -0.398 | -0.226 |
| | Students' gender | | | 0.028 | 1.046 | -0.024 | 0.080 |
| | Parents' education | | | 0.177*** | 6.139 | 0.121 | 0.234 |

**Moderated Mediation Effects Examination**

| Attention | Moderation Level | Effect Size | Boot Standard Error | Boot CI Lower Limit | Boot CI Upper Limit |
|---|---|---|---|---|---|
| | eff2 (Parental Short Video Usage Duration -SD) | 0.004 | 0.005 | -0.005 | 0.016 |
| | eff2(Parental Short Video Usage Duration) | -0.014 | 0.005 | -0.024 | -0.006 |
| | Eff (Parental Short Video Usage Duration +SD) | -0.033 | 0.009 | -0.051 | -0.017 |
| | Moderated mediation effect | -0.019 | 0.006 | -0.31 | -0.009 |

Note:* p<0.05

** p<0.01

*** p<0.001

moderating effects of parental short video usage duration. These findings delve into the subtle dynamic relationships among these variables, providing insights into the impact of short video usage on students and future developments.

The research results confirmed the negative correlation between elementary school students' short video usage and academic performance. This finding aligns with previous studies [20–22, 62], suggesting that excessive screen time can negatively impact students' academic performance. The research further solidified this relationship, emphasizing the importance of maintaining balance in digital media usage. Previous research suggests that students spending more time on screens waste valuable time that could be devoted to academics, leading excessively engaged students in electronic media to encounter difficulties in their studies [10, 20, 24, 26]. This result also resonates with Vygotsky's Social-cultural Theory proposed in 1934 in his book "Thought and Language," which asserts that learning and development are social and influenced by the surrounding environment [63]. Digital media, as a part of modern society, has profoundly impacted the learning methods and academic performance of elementary school students. Given the prevalence of digital media in contemporary education, educators and parents should closely monitor students' media usage habits to ensure they maintain appropriate digital media balance. Educators can guide students in properly using digital media and enhance their media literacy through educational programs, enabling better screen time management.

Secondly, this research revealed the mediating role of attention between short video usage and academic performance. On the contrary, the negative relationship between elementary

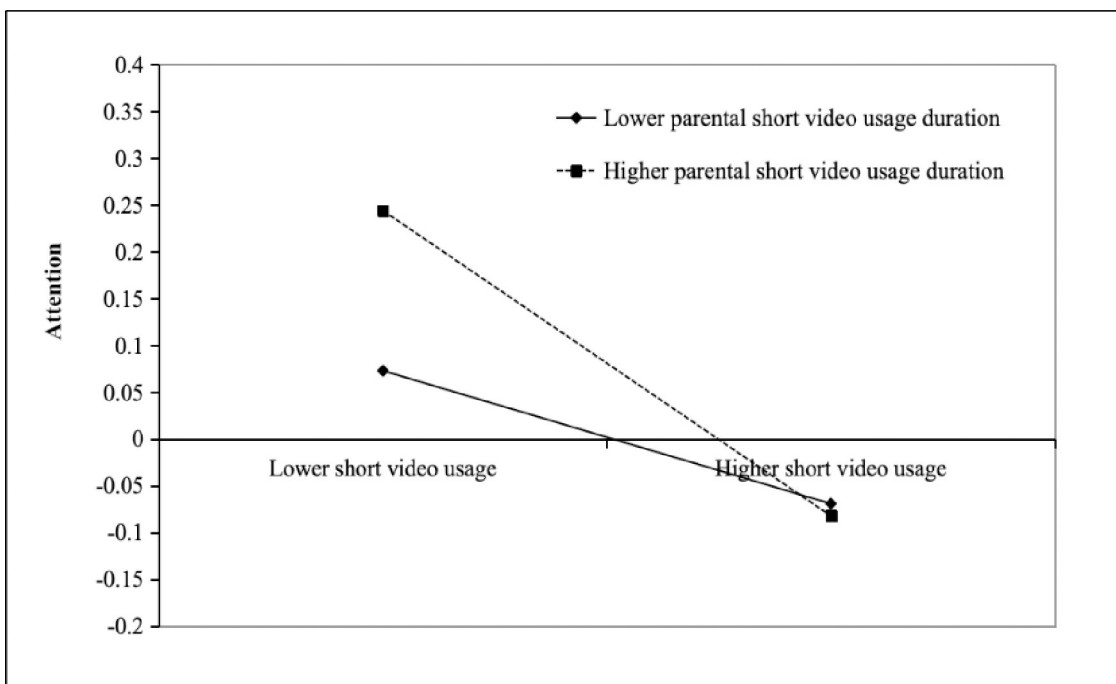

**Fig 4. The role of parental short video usage duration in the impact of short video usage on attention.**

school students' high levels of short video usage and their decreased attention has been explained in this study. Some researchers have pointed out the impact of digital media on attention span, particularly among young users, revealing that prolonged use of digital media could lead to issues like distracted attention and reduced focus [33, 35]. This research aligns with Kaplan and Kaplan's Attention Restoration Theory (ART) [34] and Lang's Limited Capacity Model of Motivated Mediated Message Processing (LC4MP) [36]. It further specifies that highly interactive engagement with short videos could weaken students' attention levels. Whereas, based on the positive relationship between attention and academic performance, when students excessively use short videos, their attention is compromised, leading to a decline in academic performance. The undeniable impact of attention issues on students' academic performance has been highlighted in previous studies [37, 38]. From a multitasking perspective, content on electronic devices often requires more frequent attention switches compared to traditional media, making it challenging for users to sustain focused attention on singular tasks like learning for extended periods [40, 42, 43]. This phenomenon reflects the complex interaction between minors' digital media habits and their cognitive functions in everyday learning environments. The impact of short video usage and attention on academic performance also aligns with the Cognitive Developmental Theory, emphasizing the joint influence of external stimuli and internal thoughts on children's cognitive abilities [64].

Thirdly, by conducting an in-depth investigation into the moderating effect of parental short video usage duration, it was found that it positively moderates the relationship between elementary students' short video usage and attention. This study emphasized that parental short video usage duration exacerbates the detrimental effects of elementary school students' short video usage on attention and further extends these adverse effects to academic performance. Previous studies have explored this issue from the perspective of parental intervention, investigating the impact of parents' active control over their children's short video usage [9, 17]. In contrast, this study is grounded in the social cognitive theory, considering the duration

of parental short video usage as an environmental and cognitive factor influencing children's behavior and attitudes [44, 46]. Integrating the Ecological Systems Theory, this study views parental short video usage duration as a part of the "microsystem," a key component within the family that plays a crucial role in the child's growth and development [15]. Specifically, parents' short video usage significantly shapes children's values and behavioral guidelines [19]. Parental short video usage demonstrates a modeling effect on children, making them perceive short video usage as acceptable behavior without generating negative attitudes or consciously exercising self-control [47–50].

Consequently, children's attention is compromised as they are allowed to weaken their attention without restrictions. Similarly, Zhang et al. [9] proposed that if parents spend excessive time on short videos, it significantly negatively predicts children's academic performance in terms of usage duration and average daily usage frequency outside of winter and summer vacations. Hence, digital media management within the family environment should be crucial in shaping children's attention and academic performance. Parents should be aware of the impact of their media usage behavior on their children and strive to establish positive patterns of short video usage. For instance, parents and children could agree to limit the time spent using short videos and adhere to it collectively, encouraging more outdoor activities and face-to-face communication. However, this study also found that in this moderated mediation model, the duration of parental short video usage positively influences children's attention, which seems counterintuitive. This might be because the duration of parental short video usage can only have its intended moderating effect when children already have a habit of using short videos. If children do not have this habit, parental short video usage duration cannot negatively impact their attention. For example, when parents use short videos, children can focus on other activities without being interrupted by their parents, thus safeguarding their attention. Regarding the positive correlation between parental short video usage duration and children's attention, we might not be able to consider it a straightforward cause-and-effect relationship. The complex reasons behind this phenomenon need further exploration in subsequent research.

### 5.2 Limitations and future research

This study explored the complex relationships between elementary school students' short video usage, attention, and academic performance. The study also examined the moderating effects of parental short video usage duration in the model. However, the study has several limitations that point to valuable future research directions.

Firstly, the study employed a questionnaire survey method to collect data. Although questionnaires are a standard data collection method, they present limitations when applied to younger students with limited literacy, vocabulary, and comprehension skills. To address this, parents assisted students in answering questions, employing techniques like reading the questions aloud and providing explanations. However, due to the sensitivity of "short video usage" within parent-child relationships, children might have refrained from being utterly truthful due to parental authority, potentially leading to self-report biases in this sample subset. Future research could consider integrating objective behavioral data and biological indicators, such as eye-tracking technology, to objectively assess the relationship between elementary school students' short video usage, attention, and academic performance.

Secondly, cross-sectional data were used in this study, which cannot capture the changes in short video usage, attention, and academic performance over time. Future research could employ longitudinal study designs to track changes in students' short video usage, attention, and academic performance over a period, revealing the dynamic relationships among them.

Additionally, long-term, in-depth interviews could be employed to qualitatively explore the psychological mechanisms underlying these relationships following children's developmental progress.

Lastly, the study's sample mainly consisted of Chinese elementary school students, limiting the generalizability of the research results to specific regions and cultures. Future research could expand the sample to include students from different countries, regions, and cultural backgrounds to understand better the impact of short video usage on students' attention and academic performance.

## 6. Conclusion

This study delved deeply into the relationships among elementary school students' short video usage, their attention, and academic performance. It also examined the moderating effects of parental short video usage duration. The research findings confirmed the close link between excessive short video usage among elementary school students and declining academic performance, highlighting the potential impact of media on students' academic performance. Furthermore, the study revealed a negative relationship between students' short video usage and their attention. The decrease in attention directly led to adverse effects on academic performance, implying a potential threat to student's cognitive abilities and prompting profound reflections on the long-term impact of digital media on children's cognitive development. Moreover, the study indicated that parents' short video usage further exacerbated the adverse effects of children's short video consumption. The research significantly expanded the knowledge domain regarding the influence of media usage on academic performance. It not only provided empirical evidence for media psychology but also offered profound insights into child mental health. The study provides practical guidance to schools and families and lays the groundwork for future research. It opens avenues for exploring innovative approaches that integrate parental education with media literacy, facilitating the translation of research findings into practical methodologies for educational practices.

## Supporting information

**S1 Raw data. Copy of raw data.**
(XLSX)

## Author Contributions

**Conceptualization:** Qiong Gong, Ting Tao.

**Formal analysis:** Qiong Gong, Ting Tao.

**Investigation:** Qiong Gong, Ting Tao.

**Methodology:** Qiong Gong, Ting Tao.

**Writing – original draft:** Qiong Gong, Ting Tao.

**Writing – review & editing:** Qiong Gong, Ting Tao.

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
