## [Decision Letter · Decision Letter 0]

7 Jul 2024

PONE-D-24-03488The relationship between Short Video Usage and academic achievement among elementary school students: the mediating effect of attention and the moderating effect of parental short video usagePLOS ONE

Dear Dr. Gao,

Thank you for submitting your manuscript to PLOS ONE. After careful consideration, we feel that it has merit but does not fully meet PLOS ONE’s publication criteria as it currently stands. Therefore, we invite you to submit a revised version of the manuscript that addresses the points raised during the review process.

We look forward to receiving your revised manuscript.

Kind regards,

Jindong Chang, Ph.D.

Academic Editor

PLOS ONE

Reviewers' comments:

Reviewer's Responses to Questions

**Comments to the Author**

1. Is the manuscript technically sound, and do the data support the conclusions?

Reviewer #1: Yes

2. Has the statistical analysis been performed appropriately and rigorously? 

Reviewer #1: Yes

3. Have the authors made all data underlying the findings in their manuscript fully available?

Reviewer #1: Yes

4. Is the manuscript presented in an intelligible fashion and written in standard English?

Reviewer #1: Yes

5. Review Comments to the Author

Reviewer #1: 1. The purpose of the study should be stated at the end of the introduction.

2. Please introduce the theoretical basis of the research hypothesis in the first section and begin to describe the research hypothesis in the second section.

3. Please complete the citation of this report "Research Report on Internet Usage among Minors in China 2021".

4. The EFA is not necessary as the scale has already been validated and it is recommended to delete Table 1.

5. The literature base for the selection of control variables should be explained.

6. in section 3.2, in addition to describing the informed consent statement, it is necessary to describe the ethical approval status.

7. As the SEM has already been carried out, it is not necessary to present a correlation analysis. I therefore propose that Table 5 be deleted.

8. Table 6 should be placed before the figure 3.

9. In the discussion it was suggested that the secondary headings should be based on the hypothetical results of the study instead of using "main findings".

10. The following literature is relevant to the study and is provided for reference:

Predicting the learning avoidance motivation, learning commitment, and silent classroom behavior of Chinese vocational college students caused by short video addiction.

The association of short video problematic use, learning engagement, and perceived learning ineffectiveness among Chinese vocational students.

The relationship between short video flow, addiction, serendipity, and achievement motivation among Chinese vocational school students: The post-epidemic era context.

Effects of short video addiction on the motivation and well-being of Chinese vocational college students.

6. PLOS authors have the option to publish the peer review history of their article (what does this mean?). If published, this will include your full peer review and any attached files.

Reviewer #1: No

---

## [Author Response · Author response to Decision Letter 0]

18 Jul 2024

Point by Point Response

I am writing this letter on behalf of my coauthor regarding our manuscript Ms. No. PONE-D-24-03488, entitled "The relationship between Short Video Usage and academic achievement among elementary school students: the mediating effect of attention and the moderating effect of parental short video usage." We want to express our appreciation to the respected editors and reviewers for providing us the constructive comments and suggestions to shape our manuscript for quality publication. According to our original reviewer's last review, we have obtained critical and rational minor suggestions to complete the manuscript's structure and content before the official publication. As you can see below, we have responded to each comment given by the reviewer corresponding to each page's revision has been made.

Reviewer #1: 

Comment 1: The purpose of the study should be stated at the end of the introduction.

Response: Thank you for your valuable comment. As suggested, we added the purpose of the study at the end of the introduction. 

Comment 2: Please introduce the theoretical basis of the research hypothesis in the first section and begin to describe the research hypothesis in the second section.

Response: Thank you for your valuable comment. We have added the information in separate headings, such as “ 2. Theoretical Basis of the Research Hypothesis” and “ 2.4. Research hypothesis”. 

Comment 3: Please complete the citation of this report "Research Report on Internet Usage among Minors in China 2021".

Response: Thank you for the comment. We added the citation of the research report “" (CYLC, CNNIC, 2023)”. 

Comment 4: The EFA is not necessary as the scale has already been validated and it is recommended to delete Table 1.

Response: Thank you for the suggestion; we deleted Table 1 with the EFA scale.

Comment 5: The literature base for the selection of control variables should be explained.

Response: Thank you for the comment. We have added the control variables, and Table 1 contains a detailed description of each control variable. 

Comment 6: in section 3.2, in addition to describing the informed consent statement, it is necessary to describe the ethical approval status.

Response: Thank you for your observation; we added the ethical approval reference under section 3.2. 

Comment 7: As the SEM has already been carried out, it is not necessary to present a correlation analysis. I therefore propose that Table 5 be deleted.

Response: Thank you for the suggestion; we deleted the table.5 with correlation analysis. 

Comment 8: Table 6 should be placed before the figure 3.

Response: Thank you for the suggestion. We have considered it, and changes have been made. 

Comment 9: In the discussion it was suggested that the secondary headings should be based on the hypothetical results of the study instead of using "main findings".

Response: Thank you for the suggestion; replace the ‘main findings’ subheading with “Exploring the Interplay of Short Video Usage, Attention, and Academic Performance among Elementary School Students: The Role of Parental Influence,” representing the study's hypothetical results. 

Comment. 10: The following literature is relevant to the study and is provided for reference:

i) Predicting the learning avoidance motivation, learning commitment, and silent classroom behavior of Chinese vocational college students caused by short video addiction.

ii) The association of short video problematic use, learning engagement, and perceived learning ineffectiveness among Chinese vocational students.

iii) The relationship between short video flow, addiction, serendipity, and achievement motivation among Chinese vocational school students: The post-epidemic era context.

iv)Effects of short video addiction on the motivation and well-being of Chinese vocational college students.

Response: Thank you for the suggestion. We added all four citations to our paper to elaborate on our literature and discussion.

---

## [Editor Report · Decision Letter 1]

21 Aug 2024

The relationship between Short Video Usage and academic achievement among elementary school students: the mediating effect of attention and the moderating effect of parental short video usage

PONE-D-24-03488R1

Dear Dr. Tao,

We’re pleased to inform you that your manuscript has been judged scientifically suitable for publication and will be formally accepted for publication once it meets all outstanding technical requirements.

Kind regards,

Jindong Chang, Ph.D.

Academic Editor

PLOS ONE
---

## [Editor Report · Acceptance letter]

16 Oct 2024

PONE-D-24-03488R1 

PLOS ONE

Dear Dr. Tao, 

I'm pleased to inform you that your manuscript has been deemed suitable for publication in PLOS ONE. Congratulations! Your manuscript is now being handed over to our production team.

Kind regards, 

on behalf of

Dr. Jindong Chang 

Academic Editor

PLOS ONE